# The Scaly-foot Snail genome and implications for the origins of biomineralised armour

Jin Sun [1,10], Chong Chen [2,10], Norio Miyamoto[2,10], Runsheng Li [3], Julia D. Sigwart [4,5], Ting Xu [3], Yanan Sun [1], Wai Chuen Wong [1], Jack C. H. Ip [3], Weipeng Zhang [1], Yi Lan[1], Dass Bissessur[6], Tomo-o Watsuji[2,7], Hiromi Kayama Watanabe[2], Yoshihiro Takaki [2], Kazuho Ikeo[8], Nobuyuki Fujii[8], Kazutoshi Yoshitake[9], Jian-Wen Qiu [3], Ken Takai [2✉] & Pei-Yuan Qian [1✉]

The Scaly-foot Snail, *Chrysomallon squamiferum*, presents a combination of biomineralised features, reminiscent of enigmatic early fossil taxa with complex shells and sclerites such as sachtids, but in a recently-diverged living species which even has iron-infused hard parts. Thus the Scaly-foot Snail is an ideal model to study the genomic mechanisms underlying the evolutionary diversification of biomineralised armour. Here, we present a high-quality whole-genome assembly and tissue-specific transcriptomic data, and show that scale and shell formation in the Scaly-foot Snail employ independent subsets of 25 highly-expressed transcription factors. Comparisons with other lophotrochozoan genomes imply that this biomineralisation toolkit is ancient, though expression patterns differ across major lineages. We suggest that the ability of lophotrochozoan lineages to generate a wide range of hard parts, exemplified by the remarkable morphological disparity in Mollusca, draws on a capacity for dynamic modification of the expression and positioning of toolkit elements across the genome.

[1] Department of Ocean Science, Division of Life Science and Hong Kong Branch of the Southern Marine Science and Engineering Guangdong Laboratory (Guanzhou), The Hong Kong University of Science and Technology, Hong Kong, China. [2] X-STAR, Japan Agency for Marine-Earth Science and Technology (JAMSTEC), 2-15 Natsushima-cho, Yokosuka, Kanagawa 237-0061, Japan. [3] Department of Biology, Hong Kong Baptist University, Hong Kong, China. [4] Marine Laboratory, Queen's University Belfast, Portaferry, N. Ireland. [5] Senckenberg Museum, Frankfurt, Germany. [6] Department for Continental Shelf, Maritime Zones Administration & Exploration, Ministry of Defence and Rodrigues, 2nd Floor, Belmont House, 12 Intendance Street, Port-Louis 11328, Mauritius. [7] Department of Food and Nutrition, Higashi-Chikushi Junior College, Kitakyusyu, Japan. [8] National Institute of Genetics, 1111 Yata, Mishima, Shizuoka, Japan. [9] Graduate School of Agricultural and Life Sciences, The University of Tokyo, 1-1-1, Yayoi, Bunkyo-ku, Tokyo, Japan. [10] These authors contributed equally: Jin Sun, Chong Chen, Norio Miyamoto. ✉email: kent@jamstec.go.jp; boqianpy@ust.hk

The appearance of biomineralised skeletons in the Cambrian precipitated an evolutionary arms race and the original explosive diversification of modern animal forms[1]. Since then, a long history of body plan modifications, reversals, and convergences has obscured the relationships among some animal lineages[2]. Understanding the genomic toolkit that enabled innovations in skeletons and armour is critical to reconstructing the early radiation of major clades[3,4]. In particular, Lophotrochozoa (including annelids, brachiopods, molluscs and others) presents a significant and unresolved problem for phylogenetic reconstruction, and controversies over the affinities of key fossils[5]. The Scaly-foot Snail, *Chrysomallon squamiferum*, is unique among gastropod molluscs in having dense, imbricating chitinous sclerites covering the whole distal surface of the soft foot[6], forming a dermal scale armour in addition to a solid calcium carbonate coiled shell (Fig. 1). These hard parts, including sclerites and the shell, are often mineralised with iron sulfide, making it the only known metazoan using iron as a significant component of skeleton construction[7].

Living hydrothermal vent taxa represent Cenozoic radiations[8] (≤66 million years ago, Mya), including peltospirid gastropods[9,10] such as the Scaly-foot Snail; the Scaly-foot Snail represents a recent evolution of a complex scleritome. On its discovery in the iron-rich Kairei hydrothermal vent field in the Indian Ocean in 2001, the complex armature of the Scaly-foot Snail was compared with apparently plesiomorphic scleritomes found in aculiferan molluscs and important early fossil taxa such as *Halkieria* and other sachtids[7]. Later, a second population lacking iron sulfide mineralisation was discovered in the Solitaire hydrothermal site[11] characterised by low concentrations of iron (1/58 of that found at Kairei[12]), which presents an opportunity to understand physiological and molecular mechanisms behind its unique mineralisation pattern (also see Supplementary Note 1). Recent results indicated that the Scaly-foot Snail mediates its scale biomineralisation by supplying sulfur through nano-scale channel-like columns in the scales, which reacts with iron ions diffusing inwards from the vent fluid to make iron sulfide nanoparticles[12]. Many sachtids also had tissue-filled canals within their sclerites,

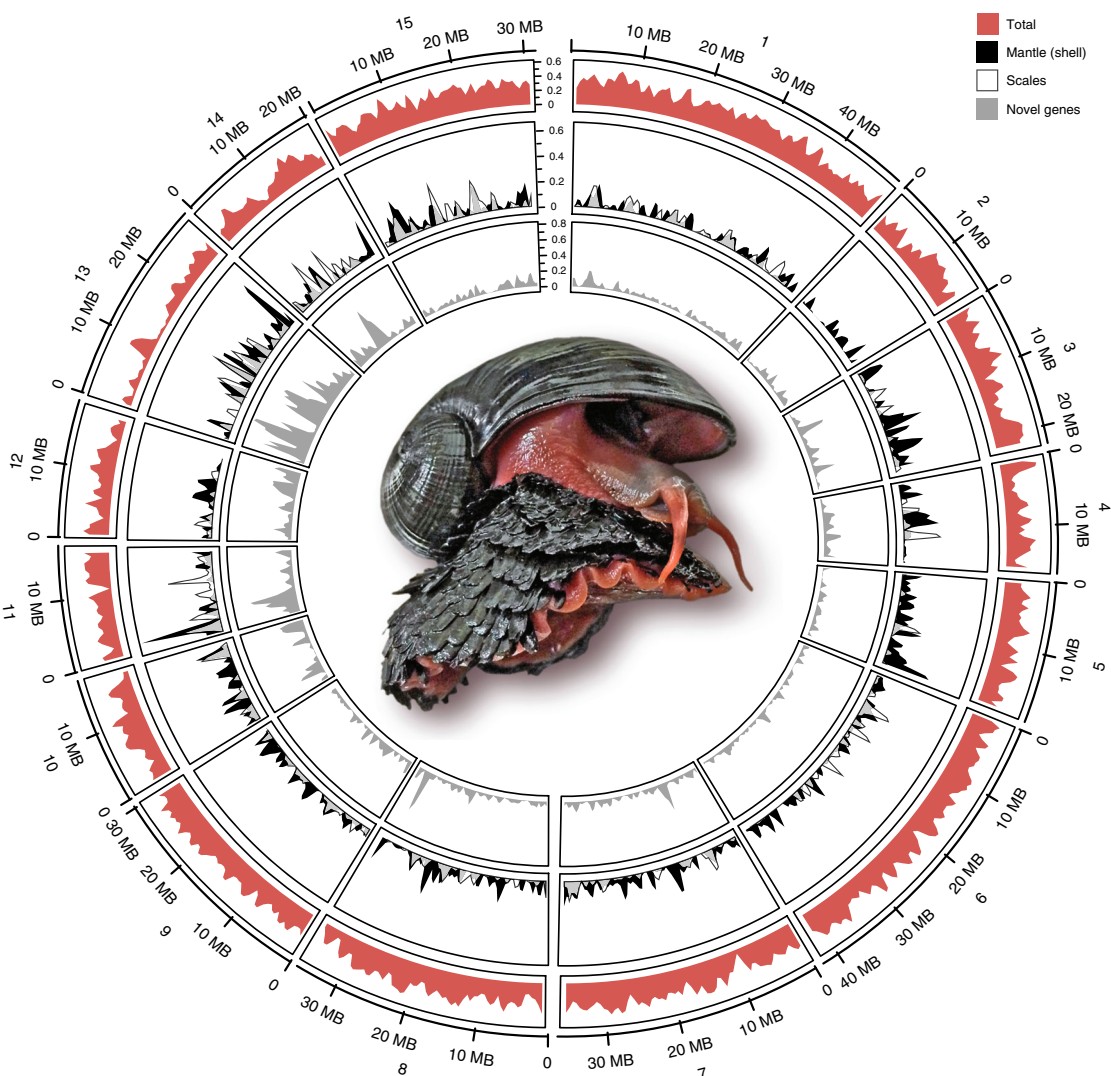

**Fig. 1 Key features of the Scaly-foot Snail *Chrysomallon squamiferum* genome.** Circos plot showing the 15 pseudo-chromosomal linkage groups; with the Scaly-foot Snail at centre. The outer ring (red peaks) indicates gene density in each pseudo-chromosome, and the inner rings shows the normalized density of the highly expressed genes in the shell-secreting mantle (black peaks) and scale-secreting epithelium (white peaks, semi-transparent overlaid on black mantle peaks) as well as the density of novel genes (grey). The expression level is the average fold change of the target tissue versus the other four types of tissues (*n* = 5); the sliding window size is 10 kb. MB, megabases. Source data are provided as a Source Data file.

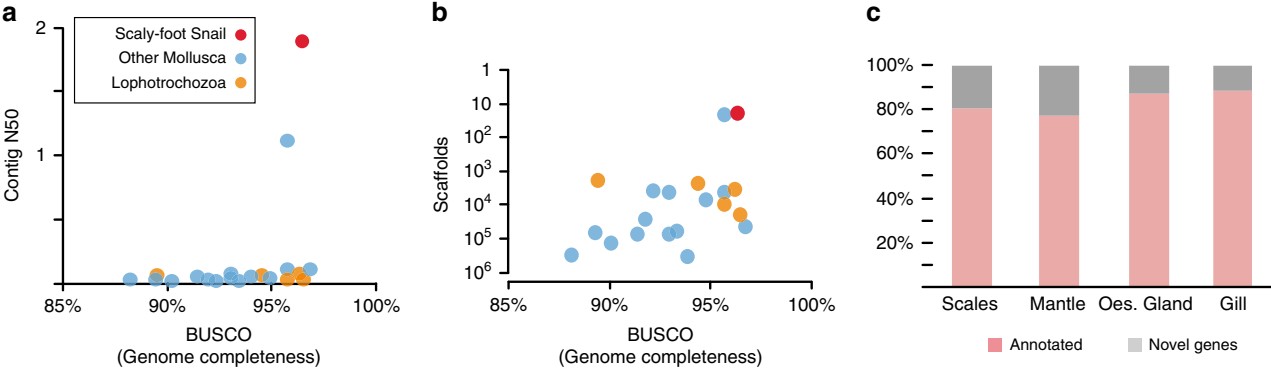

**Fig. 2 Summary of the Scaly-foot Snail genome. a, b** Quality comparisons between the Scaly-foot Snail genome and other available lophotrochozoan genomes: **a** Contig N50 vs BUSCO (Benchmarking Universal Single-Copy Orthologs); **b** Number of scaffolds vs BUSCO (for full comparison see Supplementary Table 2). Red dot indicates the Scaly-foot Snail genome, blue dots indicate other molluscan genomes, orange dots indicate other non-molluscan lophotrochozoan genomes. **c** Proportion of annotated genes (pink) and novel genes (grey) in four key tissue types of the Scaly-foot Snail, including scale-secreting epithelium, shell-secreting mantle, oesophageal gland (oes. gland), and gill. Source data are provided as a Source Data file.

although entirely different in proportion and arrangement. The channels within the scales of the Scaly-foot Snail are likely linked to another key adaptation, in that it hosts sulfur-oxidising bacteria within cells of a highly vascularised, hypertrophied oesophageal gland[13,14], and the sulfur may originate as metabolites from the endosymbionts[12]. Taking these observations together, it is unclear whether the evolution of the sclerites in the Scaly-foot Snail should be interpreted as a recurring ancestral phenome, or a recently derived adaptive novelty.

The search for a biomineralisation toolkit underlying hard part evolution in molluscs and lophotrochozoans has previously focused on molluscan mantle gene expression and shell formation[4]. Molluscs have repeatedly 'invented' additional sclerite-like hard structures, in chitons, aplacophorans, *Chrysomallon*, and also in other gastropods, cephalopods, and bivalves[15]. Here, we use a complete genome assembly of the Scaly-foot Snail and tissue-specific transcriptomics to compare with data from other lophotrochozoans in order to test whether there is indeed a universal biomineralisation toolkit in Mollusca or Lophotrochozoa, and to identify specific genomic tools that enable the Scaly-foot Snail to repeatedly modify and duplicate hard parts.

## Results

**Genome assembly**. The genome of a single specimen of *Chrysomallon squamiferum* (Fig. 1, Supplementary Fig. 1, Supplementary Table 1) collected from the Kairei hydrothermal vent field was sequenced with a combination of Oxford Nanopore Technologies and Illumina platforms (Supplementary Note 2; Supplementary Table 2). Sulfur-oxidising endosymbionts within the body[13] are a source for significant potential contamination and therefore bacterial endosymbiont genomes[16] were removed from downstream analyses (Supplementary Note 2). The genome of the Scaly-foot Snail is relatively compact for Mollusca (444.4 Mb; Supplementary Table 1) and is highly heterozygous (1.38%) but with relatively low repeat content (25.2%). With additional Hi-C data using a second specimen from the same population, 1032 contigs (N50 = 1.88 Mb) were anchored to 15 pseudo-chromosomal linkage groups (Fig. 1). A total of 16,917 gene models were predicted (85.7% comparatively annotated; 2415 additional novel genes) with evidence from transcripts, homologue proteins, and ab initio methods (Supplementary Note 2). The number of genes in the Scaly-foot Snail genome appears to be low, and the fact that 97.3% of the de novo assembled transcriptome could be mapped to the genome indicates that the genome assembly has high completeness.

The Scaly-foot Snail genome represents the most complete and continuous genome among the assembled mollusc an or lophotrochozoan genomes to date (Fig. 2), with a metazoan BUSCO (Benchmarking Universal Single-Copy Orthologs) score of 96.6% for the genome assembly and 87.5% for the predicted transcripts. This genome assembly represents a successful application of methods originally developed for model organisms[17] to a little-studied taxon, providing a benchmark in quality compared with other published lophotrochozoan genomes confounded by heterozygosity and high repeat contents (Fig. 2a, b; Supplementary Table 2).

**Gene family analyses**. The Scaly-foot Snail has fewer novel gene families than other lophotrochozoans with high-quality genomes available (such as *Pomacea canaliculata*, *Mizuhopecten yessoensis*, *Lingula anatina*, *Achatina fulica*, *Sinonovacula constricta* and *Capitella teleta*: Fig. 3a; Supplementary Table 2), which appears to conflict with an expectation that evolutionary novelties are associated with novel genes[18] (Fig. 3a). To corroborate this, only 11% of the Scaly-foot Snail gene families are not found among four other lophotrochozoan genomes, while in the other taxa, which apparently lack the dramatic morphological novelties of the Scaly-foot Snail, this figure is over 20% and up to 35% (Fig. 3a). In addition, only 4.8% of gene families in the Scaly-foot Snail genome were found to be novel to gastropods, while as much as 87.0% of them may be unique to Lophotrochozoa or even pre-date the origin of Lophotrochozoa (Fig. 3b).

Comparative analyses among available lophotrochozoan genomes (n = 15; Supplementary Table 4) showed that 351 gene families were significantly expanded in the Scaly-foot Snail. A GO enrichment analysis on the expanded gene families revealed 30 overrepresented GO categories in the Scaly-foot Snail genome (Supplementary Fig. 2, Supplementary Data 1). These expanded gene families appear to be involved in the secretion process of proteinaceous materials (e.g. scavenger receptor activity, carbohydrate binding, chitin-related metabolic process and chitin binding) and symbiosis (regulation of innate immune response and symbiont-containing vacuoles), suggesting their contributions to both biomineralisation and regulation of endosymbiosis. The expanded genes were biased in distribution across the chromosomes, with Chr11 and Chr12 being especially enriched (Fold change > 2 and FDR < 0.01, Supplementary Note 3).

We recovered the complete Antennapedia (ANTP) *Hox* gene complex containing 11 *Hox* genes on Chr11 (Fig. 4a) as in *Lottia gigantea*[19] (a gastropod) and *Mizuhopecten yessoensis*[20] (a

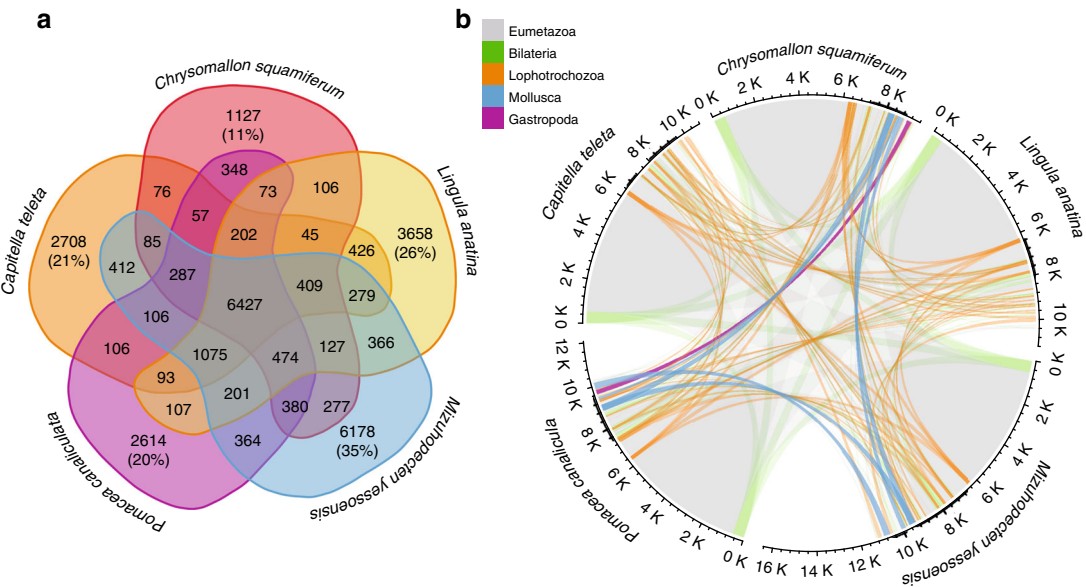

**Fig. 3 Gene family analyses of lophotrochozoan genomes. a** Venn diagram showing the number of shared and unique gene families among five lophotrochozoan genomes. **b** Circos plot showing the proportion and origin of shared gene families across five lophotrochozoan taxa. Arc values correspond to the number of gene families. Genes shared across Eumetazoa are indicated by grey lines, Bilateria by green lines, Lophotrochozoa by orange lines, Mollusca by blue lines, and Gastropoda by purple lines. K, kilo (1000). Source data are provided as a Source Data file.

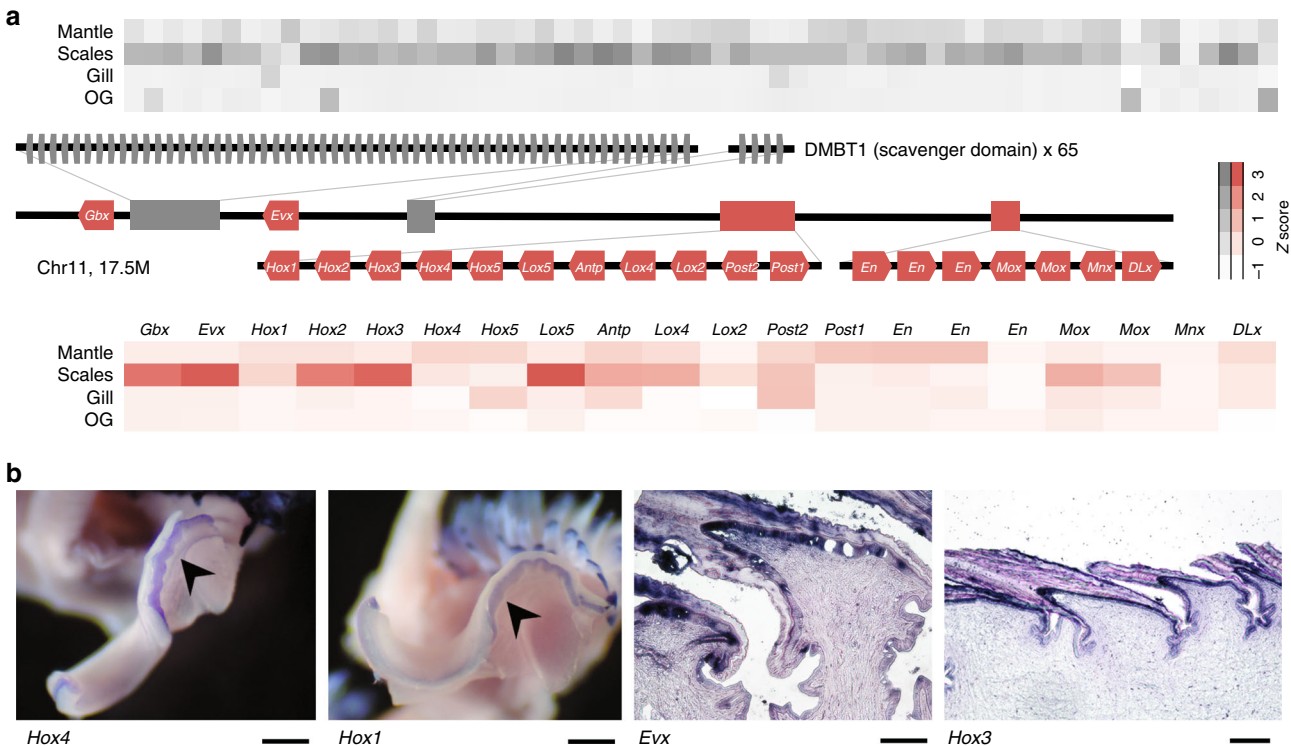

**Fig. 4 Tissue-specific gene expression in the Scaly-foot Snail. a** Arrangement of DMBT1 gene tandem repeats (grey), *Hox* genes and *Hox*-like genes (red) on the pseudo-chromosome Chr11 and their expression patterns in four types of tissues including mantle, scale-secreting epithelium, gill, oesophageal gland (OG); darker shades indicate higher levels of expression. SRCR (scavenger receptor cysteine-rich) domain, scavenger receptor cysteine-rich domain. Source data are provided as a Source Data file. **b** Tissue-specific expression patterns of transcriptome factors shown with in situ hybridisation, including *Hox4* and *Hox1* in the mantle edge (arrowheads; additional tissue is visible in the background including scales also expressing *Hox1*) and tissue sections of the scale-secreting epithelium expressing *Evx* and *Hox3*. Scale bars = 500 μm for mantle and 100 μm for scales. In situ hybridisation experiments were repeated independently on three individuals (twice on each individual for *Hox1* and *Hox3*, once for *Hox4* and *Evx*) with similar results.

bivalve); among molluscs, the intact ordered cluster has only been recovered in one gastropod (*Lottia*), and two bivalve (*Mizuhopecten*, and *Azumapecten farreri*) genomes[21], but are critical to understanding body plan evolution[22]. The derived Scaly-foot Snail homeobox genes show reorganisation by intra-chromosomal rearrangement (Supplementary Figs. 3, 4). Syntenic comparisons reveal both extensive inter- and intra-chromosomal rearrangements between the Scaly-foot Snail and other molluscs (Supplementary Fig. 3).

Gene expression levels in different tissue types of the Scaly-foot Snail were assessed by transcriptome sequencing (five specimen replicates) using samples fixed in situ in the snail's native deep-sea vent habitat[23] to avoid the confounding effect of environment and hydrostatic pressure changes on transcription. The five primary tissue types targeted included the shell-secreting mantle, scale-secreting epithelium, oesophageal gland where the bacteriocytes housing the endosymbionts are located, the gill, and the foot musculature.

Among the expanded genes on Chr11 (262 genes), one that controls protein binding to proteoglycan (for instance, chitin) known as DMBT1[24] with scavenger receptor cysteine-rich (SRCR) domains was found to be highly expressed in both the scale-secreting epithelium and shell-secreting mantle (Fig. 4a) with up to 65 tandemly duplicated paralogues compared with one or two copies in other molluscs[25]. Proteins with SRCR domains are commonly detected in shell matrix proteins[18]. These gene copies were highly expressed in the mantle, but were also especially highly expressed in the scale-secreting epithelium (Fig. 4a). The Scaly-foot Snail DMBT1 gene copies had an average of 2.3 SRCR domains, but the number varied greatly among individual paralogues (0–5; Supplementary Data 2). As SRCR domains are known to evolve rapidly to generate high gene diversity, the variation of the domain numbers in each paralogue likely reflects rapid evolution and expansion of this gene to coordinate protein secretion of dermal scales and the shell periostracum. A gene tree of the Scaly-foot Snail DMBT1 shows that the paralogues segregate into two clades, with only one copy from each clade being present in the vetigastropod *Haliotis* genome (Supplementary Fig. 5).

Molluscan shell matrix proteins are commonly rich in repetitive low-complexity domains (RLCDs)[26], which have been attributed to a number of functions such as chitin or calcium binding and structural support, but such domains are lacking in DMBT1 paralogues in the Scaly-foot Snail (Supplementary Data 3). Analyses of RLCD-rich genes across the Scaly-foot Snail genome revealed that the ratio of RLCDs in genes highly expressed in the scale-secreting epithelium is similar to the overall ratio of the whole genome, whereas a higher ratio is seen in the mantle. This indicates that RLCDs likely contribute to shell biomineralisation like known for other molluscs, but not the scales; suggesting that RLCDs may be linked with calcium carbonate biomineralisation which is lacking in the scales (Supplementary Table 3).

A further 79 gene families were found to be significantly contracted in the Scaly-foot Snail genome relative to other lophotrochozoans, including several enzymes that play critical roles in steroid and amino acid synthesis, in keeping with similar gene reductions in other chemosymbiont-dependent holobionts[23] (Supplementary Data 4). Numerous enriched GO terms (Supplementary Note 4) in highly-expressed genes in the oesophageal gland are relevant to symbiosis, including oxidation-reduction process compensating the reactive oxygen species generated by the endosymbiont[27].

**Biomineralization toolkit**. Until the early 2000s, the only living molluscs known to regularly possess imbricating sclerites were

'aculiferan' molluscs (i.e. chitons and the worm-like Caudofoveata and Solenogastres). The Scaly-foot Snail possesses an expanded suite of mineral-secreting tissues[4], in the foot epidermis as well as the shell-forming mantle tissue. Many distinctive morphological adaptations in *Chrysomallon* are connected to supporting endosymbiotic microbes housed in an enlarged oesophageal gland, a condition resulting from rapid evolution[27]. Indeed, even the scales of the Scaly-foot Snail are apparently related to supporting its endosymbionts[12].

To assess tissue-specific functions, particularly in relation to biomineralisation, a paired test in DESeq2 was applied to distinguish tissue highly expressed genes ($n = 5$, Figs. 1, 4a, and Supplementary Data 5–7), further confirmed with real-time PCR (two replicates, five tissue types, Supplementary Note 5) and in situ hybridisation (ISH; for mantle and scale-secreting epithelium; Figs. 4b, 5 and Supplementary Fig. 6). Given the high quality of the Scaly-foot Snail genome, genes related to biomineralisation can be accurately linked to their organisation in the genome and their respective expression levels in relevant tissues, i.e. shell-secreting mantle and scale-secreting epithelium.

Visualisation of expression levels by genome organisation revealed similar expression patterns for scale-secreting epithelium and mantle at chromosomal level across the genome, also revealing a pattern where the pseudo-chromosome Chr11, which contains the abovementioned ANTP *Hox* cluster as well as the DMBT genes, contained a high density of highly expressed genes in these two biomineralising tissues (Fig. 1). The expression profiles of biomineralisation genes in the scale-secreting epithelium was most similar to those of the shell-secreting mantle tissue (81 shared highly expressed genes, 11.2% of the scale-secreting epithelium or 6.8% of mantle highly expressed genes); both expressed common genes involved in biomineralisation (Supplementary Data 5, 6), in separate tissues with different and complex functions.

We identified 25 highly expressed transcription factors in the scale-secreting epithelium (12) and mantle (13), often confirmed by ISH (Figs. 4b, 6 and Supplementary Data 5, 6); the two biomineralisation tissues did not share any highly expressed transcription factors (Fig. 3c). Several of these were only detected as highly expressed in the Scaly-foot Snail hard parts among the available lophotrochozoan tissue-specific data (Fig. 3). However, all transcription factors reported to date from other lophotrochozoan skeletal elements (brachiopod mantle and chaetae, and annelid chaetae, $n = 10$ genes) were found to be also highly expressed in the Scaly-foot Snail, in either the mantle ($n = 6$ genes) or scale-secreting epithelium ($n = 4$). Some of these co-opted transcription factors, and others, have also been reported from other molluscs including aculiferan mantle and sclerite-secreting tissue, gastropod or bivalve mantle tissue, or tissues secreting the radula, beak, or operculum[21,22] (Fig. 6b and Supplementary Table 4). The involvement of these transcription factors in the formation of various hard parts across different lophotrochozoan groups ranging widely from plesiomorphic (e.g. brachiopod shell) to more recently evolved armature (e.g. Scaly-foot Snail scales) signify that they together comprise the biomineralisation toolkit preserved from the ancestral lophotrochozoan genome.

Taxon coverage of high quality lophotrochozoan genomes is currently too limited to make significant phylogenetic inferences. Internal nodes within our time calibrated phylogenetic reconstruction confirm and refine some previous divergence estimates. For example, Pteriomorphia, the best-sampled lineage within Mollusca, is recovered with a divergence estimate around 470 Mya (483.5–465.1 Mya) which corresponds to the early fossil record for the group[28]. Dating of earlier nodes (origins of Gastropoda, Bivalvia and Lophotrochozoa), and the topology, are

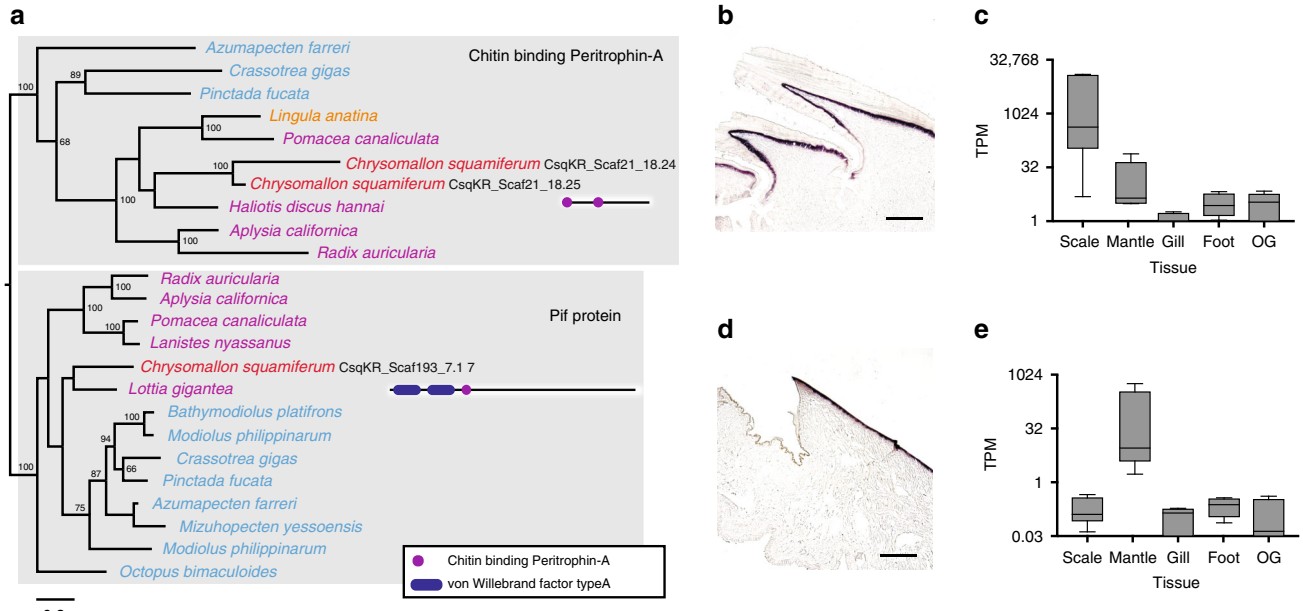

**Fig. 5 Characterisation of chitin binding peritrophin-A gene and *pif* gene in *Chrysomallon squamiferum*. a** Phylogenetic analysis of chitin binding peritrophin-A and *pif* genes with names in red indicating genes from the Scaly-foot Snail, and domain architecture of the specific genes displayed on the right of names (purple indicates chitin binding peritrophin-A and blue indicates von Willebrand factor type A). **b** In situ hybridization of chitin binding Peritrophin-A in scales. **c** Boxplot showing the expression level of chitin binding peritrophin-A in five different tissue types including mantle, scale-secreting epithelium, gill, oesophageal gland (OG), and foot. The box depicts first to third quartile with whiskers indicating maximum and minimum expression levels, and the centre line refers to the median. Source data are provided as a Source Data file. **d** In situ hybridization showing the expression of *pif* gene in mantle. **e** Boxplot showing the expression level of *pif* in five different tissues types. Within the Boxplot, the centre line refers to the median, and the boxplot depicts the first to the third quartile, with the whiskers indicating maximum and minimum expression levels.

likely confounded by limitations of the available data. Within gastropods, we recovered Neomphaliones (represented by *Chrysomallon*) sister to Vetigastropoda, and this clade sister to Patellogastropoda (Fig. 6a). One recent phylotranscriptomics study[29] similarly found Vetigastropoda sister to Patellogastropoda, but that analysis did not include any examples of Neomphaliones. Neither our genome tree nor any phylotranscriptomic analyses to date have included all gastropod subclasses as no whole genome is available for Neritimorpha[29]. Early multi-gene phylogenetic analyses repeatedly encountered severe long branch attraction artefacts particularly from patellogastropod sequences[10]. A recent mitogenome analyses including all gastropod subclasses found Patellogastropoda as the earliest-branching lineage within living gastropods[10,30], which is the topology predicted by morphology[31]. The debate on internal relationships among gastropods is likely to continue until more representatives from all subclasses become available for robust genome-based phylogeny.

Highly expressed 'novel' or taxon-specific genes were distributed across all tissue types in the Scaly-foot Snail. Although there were relatively slightly more novel genes in biomineralising tissues, the presence of novel genes was not restricted to areas of obvious morphological adaptations. There were fewer novel genes in the scale-secreting epithelium than the shell-secreting mantle, and fewer still in the symbiont-hosting oesophageal gland (Fig. 2c). Although these novel genes do not correspond to other previously annotated genes, and were newly discovered within the Scaly-foot Snail genome, they may yet be present in other taxa. This is only the first genome in the subclass Neomphaliones, and the poor taxon coverage and limited completeness of existing molluscan genomes do not provide a reliable reference framework to infer whether these 'novel' genes are truly lineage-specific

genes (i.e, Scaly-foot Snail synapomorphies), or shared with, but as yet undiscovered in, other related taxa.

Genes known to produce proteins involved in molluscan shell formation, such as *pif*, chitin-binding peritrophin-A domain gene, and chitin synthase[32], were also conserved and highly expressed in the shell-secreting mantle and scale-secreting epithelium of the Scaly-foot Snail (Fig. 5 and Supplementary Fig. 6). A GO enrichment analysis on highly expressed genes in the mantle and scale-secreting epithelium revealed that the categories integral component of membrane, scavenger receptor activity, and cell-matrix adhesion were enriched in both of these biomineralisation tissues (Supplementary Note 4), further indicating that they elements of a shared biomineralisation toolkit.

These possible downstream biomineralisation genes may be controlled by the transcription factors in the biomineralisation toolkit, and in a number of cases their specific expression in the scale-secreting epithelium were confirmed by ISH (Fig. 4b). The gene exhibiting the highest expression level among the scale-secreting epithelium highly expressed genes was chitin-binding peritrophin-A. This gene (CsqKR_Scaf21_18.25), together with its recent duplicated paralogue (CsqKR_Scaf21_18.24), were also highly expressed in the shell-secreting mantle but at lower levels than in the scale-secreting epithelium. They also show sequence similarity with the well-known shell matrix protein *pif*[33], which by contrast was highly expressed in the mantle in the Scaly-foot Snail *C. squamiferum*, similar to other Mollusca, but not in the scale-secreting epithelium. A similar pattern is seen in the chitin synthase gene family, as the Scaly-foot Snail genome possesses diverse subgroups of this family, and different paralogues were found to be highly expressed in the mantle or scale-secreting epithelium (Supplementary Fig. 6). These chitin synthase genes

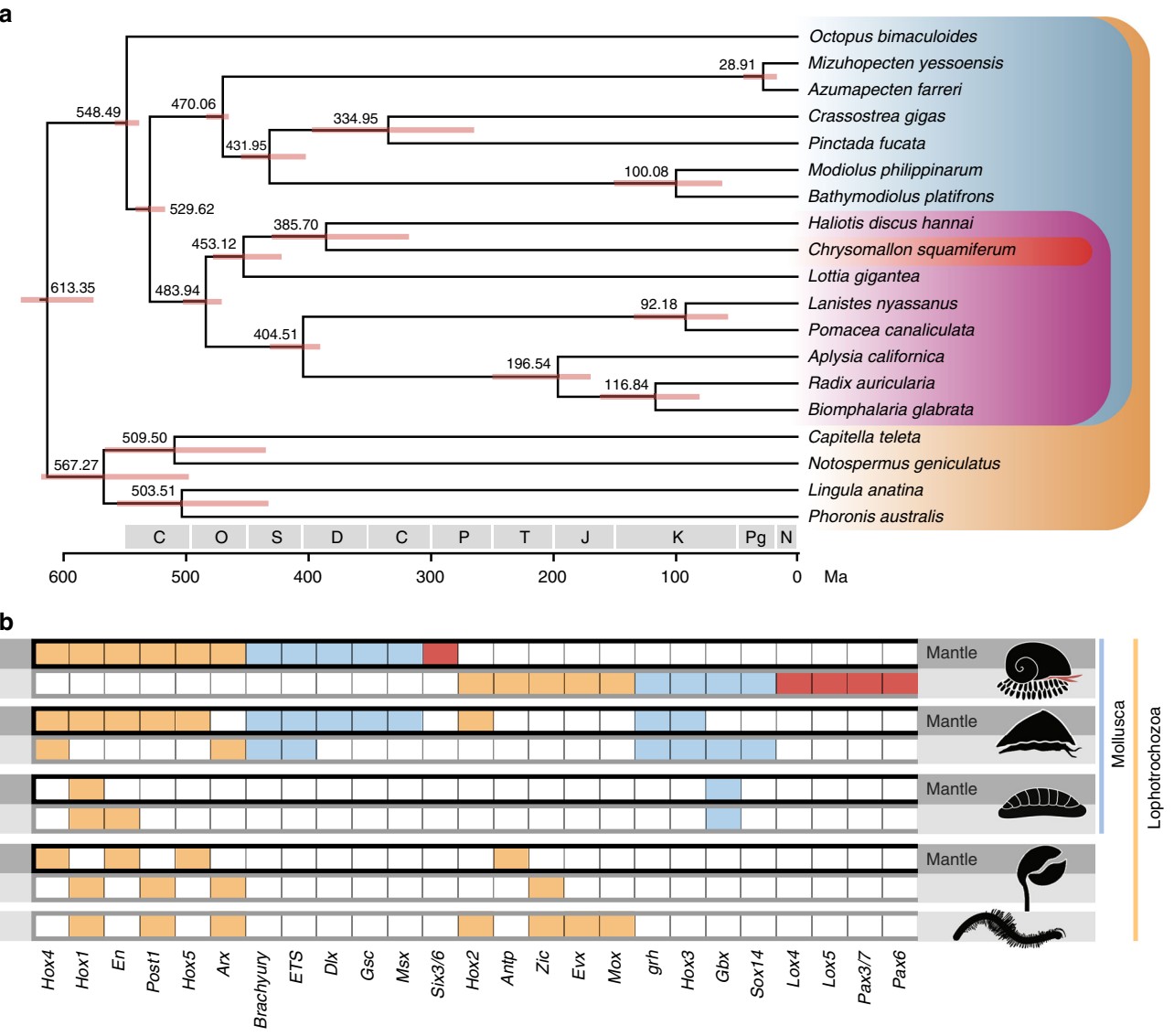

**Fig. 6 Genomic and transcriptomic comparisons across Lophotrochozoa and the biomineralisation toolkit. a** Genome-based phylogeny of selected taxa showing the position of the Scaly-foot Snail among lophotrochozoans and divergence times among molluscan lineages. Error bars indicate 95% confidence levels. The Scaly-foot Snail is highlighted in red, Gastropoda in pink, Mollusca in blue, and Lophotrochozoa in orange. **b** Transcription factors shown to be involved in armature secretion in the Scaly-foot Snail, compared with conchiferan molluscs (bivalves, gastropods, cephalopods), aculiferan molluscs (chitons), brachiopod, and annelids. Transcription factors only verified for the Scaly-foot Snail are highlighted in red, those known to be shared across Mollusca in blue, and those shared across Lophotrochozoa in orange. The top row for each group (darker shading) shows records of significant expression in shell-secreting mantle and bottom row (lighter shading) shows expression for other hard parts such as scales. Ma, million years ago. Source data are provided as a Source Data file.

are clearly involved in the biomineralisation process, and may have been co-opted in the evolution of the sclerites.

Although some highly expressed genes were different in the two biomineralising tissues, positions on the genome of transcription factors involved in the biomineralization toolkit were found to be synchronous with expression patterns seen in both mantle and scale-secreting epithelium. We propose that they control downstream novel and existing biomineralization genes in different ways to fabricate different kinds of armour or skeletal elements by upstream activation.

**Iron sulfide biomineralisation.** We compared gene expression levels in the scale-secreting epithelium and shell-secreting mantle of the Scaly-foot Snail from the iron-rich Kairei vent field with a

second population from the iron-poor Solitaire vent field that naturally lack any iron sulfide mineral coating[11]. In the Kairei population, with iron sulfide mineralisation, transmembrane transporter activity was comparatively enriched in highly expressed genes (Supplementary Data 8), supporting the hypothesis the iron sulfide precipitation is indeed mediated by the gastropod[12] through transportation and precipitation of sulfur species[12]. Another gene, metal tolerance protein (MTP) 9, exhibited over 27-fold increased expression level (Supplementary Fig. 7) in the population with iron sulfide mineralisation compared with the one without. This gene is widely found in invertebrates, protists, and even plants[34]. Although poorly studied in invertebrates, functional assays of MTP in plants revealed potential pathways for enhanced tolerance of metal ions and maintaining intracellular homoeostasis[34].

The Scaly-foot Snail biomineralises iron sulfide nanoparticles by allowing sulfur that it actively recruits or deposits in the scales to react with iron ions diffusing in from its highly iron-enriched environment[12], and the MTP9 gene likely helps the snail tolerate such high levels of iron in its surrounding environment. This would allow the Scaly-foot Snail to gain finer-scale control of nanomaterial-scale production, which is completed at much lower temperatures than can be currently controlled in laboratory settings[12]. Many other species of molluscs incorporate iron into hard parts, particularly the radula[35]. Molluscs have been highlighted as a model for generating biogenic nanomaterials at low temperatures[36], but the genomic tools to open this part of the biomineralisation toolkit for other applications were previously unavailable.

## Discussion

By comparing genomic elements and tissue-specific gene expression patterns in the Scaly-foot Snail scale- and shell-secreting tissues, as well as other biomineralising tissues in lophotrochozoans, we revealed an ancient biomineralisation toolkit comprising at least 25 transcription factors that contribute to biomineralisation across all lophotrochozoan hard parts investigated to date. Gene families in the Scaly-foot Snail genome have predominantly ancient origins, as seen in other lophotrochozoans (Fig. 3b), but their distribution and duplications across various lineages are nonsynchronous with phylogenetic positions (Fig. 3a), underlining rapid modifications. Although novel genes do appear to play important roles in downstream production of the hard parts, the hard parts themselves arise by deploying elements from the conserved biomineralisation toolkit. Comparison between sclerites from unrelated groups of molluscs —the Scaly-foot Snail, and aculiferan molluscs, or Cambrian fossils—may underline the longevity of these gene families.

The true power of the biomineralisation toolkit lies in the capacity for dynamic combination of the components being switched on or off, expanded or reduced, and relocated within the genome, which creates compounding changes in phenome. The extreme morphological disparity of mollusc biomineralisation underpins their successful diversification. Similarly, the re-use, re-arrangement and re-deployment of conserved genomic elements over more than 540 million years[37] explains both our challenges in obtaining genomic and phylogenetic resolution, and their evolutionary success.

## Methods

**Sample collection.** Specimens of the Scaly-foot Snail *Chrysomallon squamiferum* used in the present study were collected by the manned submersible *Shinkai 6500* on-board multiple deep-sea research expeditions of R/V *Yokosuka*. For genome sequencing, a specimen collected from the Kairei hydrothermal vent field (25° 19.23′S, 70°02.42′E, 2415 m depth, cruise YK16-02E, Feb 2016) and immediately placed into −80 °C upon recovery was used. For gene expression analyses, specimens fixed in situ using RNA stabilising agent from both the Kairei field (25° 19.23′S, 70°02.42′E, 2415 m depth, cruise YK13-03, Mar 2013) and the Solitaire field (19°33.41′S, 65°50.89′E, 2606 m depth, cruise YK13-02, Feb 2013) were used. Specimens for in situ hybridisation were collected at the same time from the Solitaire field, but was fixed in 4% paraformaldehyde (PFA) solution upon recovery and later transferred to 80% ethanol.

**High molecular weight DNA extraction and quantification.** The foot and mantle of a Kairei field Scaly-foot Snail (specimen code E02B1) were used for DNA extraction. High Molecular Weight (HMW) DNA was extracted using the MagAttract HMW DNA Kit (Qiagen, Hilden, Germany) according to the manufacturer's instructions. The HMW DNA was further purified and concentrated using the Genomic DNA Clean & Concentrator (gDCC-10) kit (ZYMO Research, Irvine, CA, USA). DNA quality was assessed by running 1 μl of the sample on a BioDrop μLITE (BioDrop, Holliston, MA, USA) to ensure the purified of DNA with the OD 260/280 of 1.8 and the OD 260/230 of 2.0–2.2. Concentration of DNA was assessed using the dsDNA HS assay on a Qubit fluorometer v3.0 (Thermo Fisher Scientific, Singapore). A total of 1 μg DNA was used to obtain approximate

50 Gb of Illumina Novaseq reads with the paired-end mode and a read length of 150 bp.

**Oxford Nanopore Technologies (ONT) library preparation.** A total of 2–3 μg HMW genomic DNA in 10 mM Tris-HCl (pH 8.0) were used for each library preparation. All libraries were prepared using the Ligation Sequencing Kit 1D (SQK-LSK108, ONT, Oxford, UK). The standard protocols [1D gDNA selecting for long reads (SQK-LSK108) Protocol] from Oxford Nanopore Technologies were modified and performed as follows.

The optional DNA repair step was not performed. End repair and dA-tailing were performed using the NEBNext Ultra II End-Repair/dA-tailing Module (NEB E7546). A total volume of 120 μl reaction mixture included 14 μl Ultra II End-Prep buffer, 6 μl Ultra II End-Prep enzyme mix, and 100 μl genomic DNA. The reaction mixture was incubated at 20 °C for 30 min and 65 °C for 20 min using a thermal cycler. Clean-up was performed using a 0.4× volume AMPure XP beads (48 μl), incubated at room temperature with gentle stirring for 5 min, washed twice with 200 μl fresh 70% ethanol, and briefly air-dried for 1 min to obtain the pellet. DNA was eluted by adding 31 μl of nuclease-free water (NFW), resuspending the beads, and incubating for 10 min at 37 °C. One microlitre of aliquot was quantified by Qubit to ensure that ≥1.5 μg of DNA were retained.

Ligation was then performed by gently mixing 20 μl Adaptor Mix (AMX1D, SQK-LSK108, ONT), 50 μl NEB Blunt/TA Master Mix (NEB M0367), and 30 μl dA-tailed DNA while incubating at room temperature for 30 min. The adaptor-ligated reaction was cleaned up with a 0.6 × volume (60 μl) of AMPure XP beads, incubated for 5 min at room temperature, and followed by resuspending the pellet in 500 μl Adapter Bead Binding buffer (ABB, SQK-LSK108, ONT). The purified-ligated DNA was eluted using 15 μl of Elution Buffer (ELB, SQK-LSK108, ONT), resuspending beads, and incubating for 10 min at 37 °C. One microlitre of aliquot was quantified by Qubit to ensure that ≥500 ng of DNA were retained. The aliquot of the adapted and tethered DNA (the pre-sequencing Mix) was used for loading into MinION Flow Cell.

**MinION sequencing.** MinION sequencing was performed as per manufacturer's guidelines using R9 flow cells (FLO-MIN106, ONT). Priming of the SpotON Flow Cell was preformed following the standard protocol (1D gDNA selecting for long reads (SQK-LSK108) Protocol). Directly after priming, 75 μl of the prepared library mixed with 12 μl of the pre-sequencing Mix (adapted and tethered DNA library), 25.5 μl of Library Loading Bead (LLB, ONT), 35 μl of Running Buffer Fuel Mix (RBF, ONT), and 2.5 μl of NFW were loaded through the SportON sample port in a dropwise fashion. MinION sequencing was operated with MinKNOW 1.3.23 and fastq files were base-called with Albacore v2.3.4 (ONT) with the default setting. Reads <3 kb were discarded.

**Genome assembly.** The Illumina reads were trimmed with Trimmomatic v0.33[38]. They were further assembled by Platanus v1.2.4 using the following settings: –k 31 –u 0.2 –t 29 –s 10. The genome size was estimated to be 444.4 Mb using the 17-mer histogram generated (Supplementary Note 1). The histogram was also submitted to GenomeScope (http://qb.cshl.edu/genomescope/); and the genome heterozygosity was estimated to be 1.38%. This indicates that the *C. squamiferum* genome is relatively compact in Mollusca and also highly heterozygous. The total size, N50, and mean length of the assembled genome from Platanus were 469.0 Mb, 18.2 kb and 1.8 kb, respectively, indicating high fragmentation.

We then used a range of bioinformatics pipelines to assemble the genome with ONT reads, including the ONT-only approaches (i.e. [canu version 1.7[39]], [canu version 1.7 + smartdenovo (https://github.com/ruanjue/smartdenovo)[40], and [minimap2 version 2.17-r943-dirty + miniasm version 0.3-r179], etc.) and the Illumina + ONT hybrid approach (e.g. MaSuRCA version 3.2.6). The detailed codes and settings of each assembly pipeline are detailed in Supplementary Note 2.

Following comparison of assembly statistics of different pipelines (Supplementary Note 2), the genome assembled from the canu + smartdenovo pipeline was the best one and therefore used for downstream analyses. The assembly was firstly error corrected five times with the ONT fastq file by Racon v1.4[41] and sequentially corrected twice with Illumina reads using Pilon v1.13[42]. The resultant error-corrected assembled genome had a total size, N50 and mean size of 407.8 Mb, 1.91 Mb and 393.7 kb, respectively. Analysis of the genome assembly completeness with metazoan Benchmarking Universal Single-Copy Orthologs (BUSCO) v3.0[43] revealed that the completeness of the genome is 96.6%, only 2.5% of the reported metazoan genes were missing, and confirmed the presence of single-copy metazoan genes.

**Microbial sequence contamination removal.** Genome binning was performed by using MetaBAT 2[44] with the assembled contigs as input to check the microbial sequence contamination. The resulting output genomes were examined for completeness and potential contamination using CheckM v1.0.7[45], based on the presence of particular marker gens. Open reading frames (ORFs) were predicted using Prodigal v2.6.3[46] and only ORFs with closed end were retained. Phylogenetic analysis was performed based on 16 bacterial single-copy marker genes (*frr*, *rplB*, *rplC*, *rplD*, *rplE*, *rplF*, *rplN*, *rplS*, *rpmA*, *rpsB*, *rpsC*, *rpsE*, *rpsJ*, *rpsS*, *smpB* and *tsf*) share by all the genomes included for analyses. Protein sequence of the 16 genes

were aligned individually by ClustalW in MEGA 6[47], and then linked together to construct a maximum likelihood tree. For HGT analysis, all predicted protein sequences from the two symbionts were BLASTP searched against the NCBI NR database (https://www.ncbi.nlm.nih.gov/refseq/) using an e-value 1e−5 with 50 best hits. The BLASTP output files were queried into MEGAN 6[48] for taxonomy analysis in a latent class analysis (LCA) model. Summary of removed microbial sequences can be found in Supplementary Note 2.

**Hi-C sequencing and genome scaffolding.** Another individual of the Scaly-foot Snail from the same locality, the Kairei field, and the same collection lot (specimen code E02B2) was used for Hi-C library preparation. For Hi-C library preparation from the *Chrysomallon squamiferum* specimen E02B2 from Kairei field, the following methods were used, modified from Lieberman-Aiden et al.[49]. Approximately 2 g wet weight of foot tissue stored in −80 ℃ freezer was thawed on ice and resuspended with 37% formaldehyde in serum-free DMEM for animal chromatin cross-linked. Then, the suspended tissues were homogenized and incubated at room temperature (RT) for 5 min, and glycine was added to a final concentration of 0.25 M. The solution was incubated at RT for another 5 min and transferred on ice for 15 min. The cells were further lysed in a pre-chilled lysis buffer which includes 10 mM NaCl, 0.2% IGEPAL® CA-630 (Sigma-Aldrich), 1X protease inhibitor in 10 mM PH = 8.0 Tris-HCL buffer. The chromatin digestion, labelling and ligation with biotin steps followed Lieberman-Aiden et al.[49]. The protein and biotinylated free-ends were removed, and DNA was purified and sequenced by Illumina Novaseq platform with the paired-end mode and a read length of 150 bp.

The Hi-C raw Illumina reads were trimmed with Trimmomatic v0.33[38] and then assessed by HiC-Pro v2.10.0[50]. Only the valid reads generated by HiC-Pro were further processed by the Juicer 1.5.6 pipeline[51]. Then, genomic scaffolding was conducted with the 3D de novo assembly pipeline[52] using the default diploid parameters. Several manual corrections were done in Juicebox[51] to ensure the scaffolds within the same pseudo-chromosomal linkage groups met the Hi-C linkage characteristics. In this process, three contigs were split into two parts and anchored to different chromosomes. In total, 1025 contigs were scaffolded into 15 pseudo-chromosomal linkage groups, and only seven contigs were not anchored due to insufficient Hi-C linkage found on them. The *Hox* gene cluster found in two contigs in the pre-Hi-C version was linked into an intact one, suggesting a good performance of the Hi-C scaffolding method. The final genome assembly statistics can be found in Supplementary Table 1.

**Genome quality check and repeats identification.** Quast v4.0 was used to check genome assembly quality with ONT reads, for assembly assessment report see Supplementary Note 2. Repeats and transposable elements were annotated using the RepeatModeler 2.0 and RepeatMasker 4.0.8 pipeline[53] with the searching programme of NCBI RMblast v2.9.0. The species-specific repeat library was annotated with RepeatModeler. Afterwards, RepeatMasker was run twice, one using the species-specific repeat library and another one using the repeats in Repbase (https://www.girinst.org/repbase/). All the results were summarized and classified with the perl script buildSummary.pl in the RepeatMasker package, for summary table see Supplementary Note 2.

**Genome annotations.** Two versions of transcriptome assemblies were generated: (1) the transcriptome reads (see Transcriptome sequencing section below) were de novo assembled by Trinity v2.6.5; (2) the transcriptome reads were aligned to the genome using histat2 v2.1.0[54], and the aligned .bam file was assembled by Trinity under the genome-guided mode. These two versions of transcriptome were merged by PASA pipeline v2.2.0 with the aligners of BLAT and gmap[55] and were further clustered with cd-hit-est v4.6 with a minimum sequence identity of 0.95[56].

Maker v3.0[57] was initially ran with the transcript evidence alone, and only gene models with an AED score <0.01 were retained. Gene models with <3 exons, with incomplete open reading frame, and with an inter-genic region <3 kb were removed. The rest of bona fide gene models were used to train Augustus v3.1, a de novo gene predictor[58]. Then, the gene model prediction was performed by using Maker again, but with transcript evidence, protein evidence, Augustus gene predictions, as well as an automatic annotation integration of these data into a consensus annotation based on their evidence-based weights.

Gene functions were determined by searching the NCBI non-redundant database (https://www.ncbi.nlm.nih.gov/refseq/) and SwissProt database via UniProt (https://www.uniprot.org/) with the settings of: -evalue 1e-5 -word_size 3- num_alignments 20 -max_hsps 20. The Gene Ontology (GO) functional categories were deduced from the BLAST2GO pro v4.1.9[59] software. Kyoto Encyclopedia of Genes and Genomes (KEGG)[60] annotation was performed using the KEGG Automatic Annotation Server (https://www.genome.jp/kegg/kaas/) with the bi-directional best hit (BBH) method. A sensitive HMM scanning method on the known pfam functional domains with an e-value of 0.05 was also used to classify the gene families. The low complexity protein was predicted with online XSTREAM v1.73 (https://amnewmanlab.stanford.edu/xstream/).

**Phylogenetic analyses.** The orthologue groups (OGs) were determined by a BLASTp search against protein sequences of other available high-quality molluscan, lophotrochozoan, and metazoan genomes (see Supplementary Note 6). The

BLASTp results were used to assign the OGs by OrthoMCL v2.0.9 pipeline[61]. OGs from selected lophotrochozoan taxa (Fig. 6a) were used for the phylogenomic analysis. Only single-copy genes in each OG and genes that can be found in at least 60% of taxa (i.e. at least 11 species) were retained for downstream phylogenomic analysis, resulting in 1375 OGs. Gene sequences within each OGs were aligned by MUSCLE, and the bona fide alignments were kept after trimming by TrimAL[62]. These alignments were concatenated, and the total alignments which contained missing sequences included 435,071 distinct alignment patterns across 19 species. The phylogenetic tree was conducted by RAxML v8.2.4[63] with the partition information of each orthologue gene and GTR + gamma model. The final tree file was viewed by FigTree v1.4.3 (http://tree.bio.ed.ac.uk/software/figtree/). All boot-strap values were 100, indicating full support.

MCMCtree[64], part of Phylogenetic Analysis by Maximum Likelihood (PAML) v4.9 (http://abacus.gene.ucl.ac.uk/software/paml.html), was used to predict the divergence time among molluscs, with nodes constrained by fossil records and geographic events[65]. The following fossil records and geographic events were used to constrain the nodes in the MCMC tree: A hard max time-point of 150 Ma for *L. nyassanus* and *P. canaliculata*, which correspond to the split of South America and Africa[66]; minimum = 168.6 Ma and soft maximum = 473.4 Ma for *A. californica* and *B. glabrata*[67]; hard minimum bound = 390 Ma for Caenogastropoda and Heterobranchia[68]; minimum = 470.2 Ma and soft maximum = 531.5 Ma for *A. californica* (or *B. glabrata*) and *L. gigantea*[67]; and minimum = 532 Ma and soft maximum = 549 Ma for the first appearance of molluscs[65]; hard minimum = 465.0 Ma for the first appearance of Pteriomorpha[10]; and minimum = 550.25 Ma and soft maximum = 636.1 Ma for the first appearance of Lophotrochozoan[65]. The best protein substitution model was LG + I + G and was employed to each site. The burn-in, sample frequency, and number of samples were set as 10,000,000, 1000, and 10,000, respectively.

**Gene family analyses.** The OGs deduced from OrthoMCL v2.0.9 were also used for the gene family expansion and contraction analysis by CAFÉ v3.1 pipeline[69,70]. Only those with gene family wide *P* value <0.01 and a taxon-specific Viterbi *P* value <0.01 were considered as an event of expansion/contraction.

To determine the gene age of the OGs, the OrthoMCL result was also used to deduce the time of origin, after removing taxon-specific OGs. OGs with genes only found in gastropods were classified as Gastropoda-specific. The similar approach was applied to assign OGs as Mollusca-specific, Lophotrochozoa-specific, Bilaterian-specific and Eumetazoan-specific. A binary (present/absent) matrix was also deduced from the OrthoMCL results, and taxon-specific OGs were further excluded. The binary matrix was used to plot a correlation matrix heatmap using the corrplot library in R 3.5.2[71].

In order to examine the chromosomal distribution of the expanded genes or the genes that are highly expressed in each tissue, a hypergeometric test was applied to assess whether there was any bias in any particular chromosome, with the assumption that genes are randomly distributed in each chromosome. The distribution in the expanded gene families is summarised in Supplementary Note 3.

**Transcriptome sequencing.** The Scaly-foot Snail used in genome sequencing (specimen code E02B1) was also dissected into a number of tissues/organs, namely digestive gland, scales, foot muscle, ctenidium (gill), mantle, nerve cord, nephridium, endosymbiont-containing oesophageal gland, testis and ventricle, following a previously published account of the Scaly-foot Snail anatomy[14]. This dataset was only used for gene prediction, as the sample was not fixed in situ. Meanwhile, for comparative purposes, only in situ fixed samples were considered; two individuals from the Kairei field and another three from the Solitaire field were used, targeting tissues of interest including scales, foot, gill, mantle and oesophageal gland.

RNA was extracted using TRIzol reagents and further sequenced using the Illumina Novaseq platform with the approximate output of 5 Gb for each tissue, read length of 150 bp and paired-end mode. The raw reads were cleaned with Trimmomatic v0.36[38]. Gene expression level in each tissue was quantified by Kallisto v0.44.0[72] with sequence based bias correction. Differentially expressed genes were determined by DESeq2 using the normalization method of Loess, a minimum read count of 10, and paired test (*n* = 5).

Tissue-specific genes for the tissues were determined based on their expression levels compared across all tissue types. To minimise the batch effect from pooling different sample collection events and localities, a paired test method was applied. Only genes that were overexpressed with a fold change of over 2 and FDR < 0.05 against other tissue types were classified as highly expressed. Differentially expressed genes in the scales and mantle between the Kairei and Solitaire fields were also compared with shed light on the iron sulfide biomineralisation. Dominant functions of these target genes were further assessed with GO enrichment analyses using GOEAST v.1.30[73] or in the BLAST2GO v.4.1.9[59] package (see Supplementary Note 4 for results).

**Real-time PCR validation.** Real-time PCR was employed to validate gene expression patterns in the two main tissues of interest, scales and mantle, as well as three selected other tissue types for comparison, including foot muscle, oesophageal gland, and gill. The primers for real-time PCR were designed with the on-line NCBI Primer-BLAST tool (for a list of primers see Supplementary Note 5). PCR

product length was set within the range of 100–200 bp, and optimal melting temperature was set as 60.0 °C. Only Primer-pair with the least possibilities of self-complementarity and self 3′ complementarity was selected for each gene. The elongation factor 1-alpha (EF1a) was selected as the internal standard gene, as its expression level remain almost constant across all the samples.

Total RNA was extracted from each type of tissue by Trizol method and trace amount of DNA was removed with TURBO DNA-free kit (Thermo Fisher Scientific), and the first strand cDNA was synthesized by using High Capacity cDNA Reverse Transcription Kit (Applied Biosystems). Real-Time PCR was performed with SYBR® Green RT-PCR Reagents Kit (Applied Biosystems) on LightCycler 480 II (Roche) with the following procedures: (1) polymerase activation at 95 °C for 10 min, and (2) annealing and extending at 57 °C for 1 min with a total of 40 cycles. The specificity of primer pairs for the PCR amplification was checked by the melting curve method. Triplicates were applied for each gene, and the relative gene expression level was calculated based on the $2^{\Delta\Delta Ct}$ method[74].

**In situ hybridisation**. To localize the expression of genes involved in scale and mantle formation, in situ hybridisation was performed on scale-secreting epidermis and the mantle of Scaly-foot Snails collected from the Solitaire field, fixed in 4% PFA solution and stored in 80% ethanol. In situ hybridisation was performed according to methods detailed in Miyamoto et al.[75] with slight modifications, detailed as follows.

Whole-mount in situ hybridisation was carried out for the mantle. Samples were rehydrated and washed with PBST (i.e. PBS containing 0.1% Tween 20) three times. The samples were digested with 10 μg/ml proteinase K/PBST for 30 min at 37 °C. After a brief wash with PBST, the samples were post-fixed in 4% PFA/PBST for 10 min RT (20–25 °C) and washed three times in PBST. Samples were prehybridised in a prehybridisation solution (50% formamide, 5 × SSC, 5 × Denhardt's solution, 100 μg/ml yeast RNA, 0.1% Tween 20) at 60 °C for 4 h and then hybridised with a hybridisation solution containing a digoxigenin (DIG)-labelled RNA probe at 60 °C, for 3 days. Hybridised samples were washed twice in a solution of 50% formamide, 4 × SSC, and 0.1% Tween 20 for 30 min each; then twice in 50% formamide, 2 × SSC, and 0.1% Tween 20 for 30 min twice; 2 × SSC, and 0.1% Tween 20 for 30 min each time; and twice in 0.2 × SSC and 0.1% Tween 20 for 30 min each, at 60 °C. These were then washed with MABT (i.e. maleic acid buffer containing 0.1% Tween 20) three times for 30 min at RT, blocked in 2% blocking reagent (Roche) in MABT for 2 h at RT, and incubated overnight with a 1:1500 dilution of antDIG-AP antibody (Roche) in the blocking buffer at 4 °C. Samples were then further washed six times with MABT for 60 min each on a rocker and then transferred into TNT buffer (100 mM Tris pH 9.5, 100 mM NaCl, 0.1% Tween 20). A chromogenic reaction was performed using nitro blue tetrazolium chloride/5-bromo-4-chloro-3-indolyl-phosphate (NBT/BCIP; Roche) in AP buffer (100 mM Tris pH 9.5, 100 mM NaCl, 50 mM $MgCl_2$, 0.1% Tween 20 and 2% polyvinyl alcohol) until signals were visible. The reaction was stopped in PBST, post-fixed in 10% formalin/PBST, rewashed with PBST, mounted with 40% glycerol, and observed under a light microscope (IX71, Olympus).

In situ hybridisation of the scale-secreting epidermis was carried out with sections. Samples were washed three times with PBS and mounted in Tissue-Tek O. C.T. compound to use in frozen sectioning. Frozen sections were air-dried, washed with PBST, and fixed in 4% PFA/PBS for 10 min at RT. The slides were washed with PBS and digested with 1 μg/mL proteinase K in PBS for 10 min at RT. After a brief wash with PBS, the samples were post-fixed in 4% PFA/PBS for 10 min at RT. The slides were washed with PBS three times. The samples were prehybridized for at least 1 h in prehybridisation solution (50% formamide, 5 × SSC, 5 × Denhardt's solution, 50 μg/mL yeast RNA) at 60 °C and hybridized with a DIG-labelled RNA probe at 60 °C for 3 days. The slides were washed with a solution of 50% formamide and 2 × SSC for 30 min; twice in 2 × SSC for 30 min each; 0.2 × SSC for 30 min twice at 60 °C. They were further rinsed three times with MAB, blocked in 2% blocking reagent/MAB for 2 h at RT and incubated overnight at 4 °C with a 1:1500 dilution of anti-DIG-AP antibody (Roche) in blocking buffer. Finally, they were washed six times with MAB for 30 min each and transferred into AP buffer (100 mM Tris pH 9.5, 100 mM NaCl, 50 mM $MgCl_2$ and 2% polyvinyl alcohol). A chromogenic reaction was performed using NBT/BCIP in AP buffer until a signal was visible. The reaction was stopped in PBS, postfixed in 4% PFA/PBS, washed with PBS and mounted with 80% glycerol. The hybridised samples were observed under a light microscope (IX71, Olympus).

**Reporting summary**. Further information on research design is available in the Nature Research Reporting Summary linked to this article.

## Data availability

The *Chrysomallon squamiferum* genome that support the findings of this study have been deposited in the NCBI Sequence Read Archive under the BioProject number PRJNA523462, all raw sequencing data, including Illumina and Nanopore reads, are also deposited under the same BioProject number. The assembled genome, transcriptome, predicted transcripts, proteins have been deposited in Dryad[76]. Publicly available datasets used in the study include the following: NCBI NR database (https://www.ncbi.nlm.nih.gov/refseq/), Repbase (https://www.girinst.org/repbase/), SwissProt database via UniProt (https://www.uniprot.org/), and KEGG (https://www.genome.jp/kegg/). The source data

underlying Figs. 1, 2c, 3b, 4a, 5c and 6b, and Supplementary Figs. 1, 2, 3, 6b, 7c, 8, 9, 10, 11, 12 and 13 are provided as a Source Data file. Although specimens of *Chrysomallon squamiferum* are available from the authors at a reasonable request, the numbers available are very limited.

## Code availability

All codes, commands, and intermediate files for the bioinformatics analyses carried out (using freely or commercially available software, as listed in the Methods section) in the present study are contained in Supplementary Data 9.

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

## Acknowledgements

The authors thank the captain and crew of R/V *Yokosuka* during cruises YK13-02 (principal scientist: Manabu Nishizawa), YK13-03 (principal scientist: Kentaro Nakamura) and YK16-02E (principal scientist: Ken Takai), and pilots of DSV *Shinkai 6500*. Takuro Nunoura (JAMSTEC) is gratefully acknowledged for bringing together collaborative efforts. This research was financially supported by the China Ocean Mineral Resource Research and Development Association (DY135-E2-1-03 to P.-Y.Q.), the Hong Kong Branch of South Marine Science and Engineering Guangdong Laboratory (SMSEGL20Sc01 to P.-Y.Q.), the Research Grants Council of Hong Kong (GRF grant No. 16101219 to J.S., C.C., and P.-Y.Q.), and a Japan Society for the Promotion of Science Grant-in-Aid for Scientific Research (18K06401 to C.C. and H.K.W.). Illumina sequencing was performed by Novogene (Beijing, China). Sampling in the Mauritian EEZ was approved by Ministry of Foreign Affairs, Regional Integration, and International Trade, Mauritian Government (Ref. 29/2014; 50/38/24 V2).

## Author contributions

P.-Y.Q., K.T., C.C., J.S., Y.T., J.-W.Q. and J.D.S. conceived and designed the project. C.C., T.W., H.K.W., K.T., J.D.S., K.I., N.F., K.Y., D.B. and N.M. collected and fixed the samples. C.C., J.S., T.W. and N.M. dissected the samples. J.S., Y.S., T.X. and Y.L. extracted RNA, DNA, and performed the ONT sequencing. W.Z. checked bacteria contamination. N.M. performed in situ hybridisation. R.L. and J.S. performed synteny analysis. J.S. and J.C.H.I. performed gene family analyses. J.S. assembled the genome, annotated the genes, and performed bioinformatics analyses. W.C.W. performed the real-time PCR. C.C., J.D.S.,

J.S. and N.M. interpreted the data and drafted the paper. All authors contributed to the final paper.

## Competing interests

The authors declare no competing interests.
