## [Peer Review File · Nature Communications]

Reviewers' comments:

Reviewer #1 (Remarks to the Author):

In this study, Sun et al. report the genome of the Scaly-foot Snail *Chrysomallon squamiferum* and explore the biomineralization toolkit of this unusual mollusc with a shell as well as sclerites that may be mineralized with iron sulphide. The genome is of very high quality (chromosome-level scaffolds) and will undoubtedly be a valuable resource to the molluscan genomics research community. The paper is generally well-written (but see minor comments below) and the findings are very interesting with respect to the study of molluscan lophotrochozoan biomineralization. I feel that this paper should be published but that some revision may be necessary. I found the paper to somewhat superficial with respect to some interesting findings. Of course this is somewhat unavoidable given the short format, but I feel that the organization could be improved to emphasize key findings related to the central question of the evolution of molluscan biomineralization raised in the summary. For example, I question the significance of the reported Hox gene cluster rearrangement to the formation of novel hard parts in this lineage, especially given the absence of even a hypothesized mechanism. Perhaps the reporting of these results and the relevant discussion would best be moved to a supplementary note? Additionally, I felt that the discussion of the results of the CAFÉ analysis and the GO analysis, which in my opinion have great potential to address the central question raised in the summary, was underdeveloped. I would encourage the authors to reconsider the organization and 'real estate' of the manuscript devoted to the central message(s) of the paper.

I also have a number of other minor suggested edits:

Line 38: "biosynthesis of iron" – iron in an element; animals synthesize iron compounds but not iron itself.

Line 43: I would insert "the" before "biomineralization"

Line 61 and elsewhere: Lophotrochozoa is a clade / grouping so there is no need to put the word "the" before it.

Line 62: Correct "present" to "presents"

Line 84: This is not the only ONT + Illumina genome for a non-model organism (so this is not a "novel" approach).

Line 88: Correct "among" to "of a"

Line 89: Something seems wrong here: "...complete ANTP of 11 Hox genes..."

Line 91 and elsewhere (including figures and tables): *Patinopecten* has been changed to *Mizuhopecten*.

Lines 93-95: Maybe? What would the speculated mechanism be?

Line 120: I suggest deleting "the" in "using the samples"

Lines 142-143: No genes have been reported from the spicules (I think the term sclerite should be used over spicule, by the way), beak, or operculum – these are non-living structures – but the tissues that secrete them.

Lines 151-152: It is not made clear until this part of the manuscript that the scales of animals from environments lacking iron sulfide have a different composition and the compositional difference is never made clear.

Figure 2b: The different parts of this portion of Figure 2 would be easier to interpret if split into more lettered parts with descriptive labels in the figure caption.

Figure 3: Please clarify which genes are being compared in part b of Figure 3 in the main text or figure caption. Also unless I missed it, 3b is referenced in the main text before 3a.

Methods:

It seems like OrthoMCL was used to infer orthology but the methods only briefly refer to this program. Please clarify that OrthoMCL was used and not just BLASTP. If OrthoMCL was used, the inflation parameter and all-versus-all BLASTP settings used should be provided. I would also strongly encourage the authors to make all of the output fasta files generated by OrthoMCL available as supplementary data.

Line 263: The header inadequately described the text in this section.

Line 281: Correct "preformed" to "performed"

Line 336: Delete "for"

Lines 336-337: Why just two rounds of Pilon? Were the pre-Pilon, post-round 1, and post-round 2 Pilon corrections compared?

Line 340: Potentially interesting but definitely not needed: Were the missing BUSCO genes searched for in the genome by BLAST? Are the genes missing in this genome the same ones missing in other gastropod / molluscan genomes?

Lines 356 and 359: I believe the suite is called Juicer and the tool used is called Juicebox.

Line 385: Correct "bona-fide" to "bona fide" (italicized).

Line 413: Correct "divergent" to "divergence"

Genome annotations section: Details provided on genome annotation in Maker are inadequate.

MinION sequencing section: How many flowcells were sequenced? The authors might also consider providing details on how much data was generated per flowcell in the supplement.

Supplement: I appreciate that the exact commands used with some of the various bioinformatic tools employed in this study are provided. I would encourage the authors to not hesitate to paste in the exact commands used for more of the tools they employed to help make their assemblies / analyses even more transparent. I would also encourage the authors to make key input files, scripts / commands used, and key intermediate output files from analyses available as supplementary data, especially for the evolutionary analyses that may be of interest to workers studying other areas of animal genomics (e.g., the OrthoMCL analysis result fasta files).

Data availability: More information needs to be provided in this section. It is unclear what will be made publicly available. I am of the opinion that the raw sequencing data, assembled genome, assembled transcriptomes, and all predicted transcripts and protein sequences should be made available at the time of publication of this study.

Reviewer #2 (Remarks to the Author):

Sun et al. present an important genomic analysis of one of the most novel taxa found at deep-sea hydrothermal vents. That alone would be worthy of publication. However, the presence of multiple biomineralization sites on the snail as well as the incorporation of iron into the foot scales makes this paper especially important. Their treatment and methodology appears both rigorous and comprehensive. And while of substantial interest to biologists and perhaps paleontologists, the authors also touch on important considerations and insights in bio-engineering and the possible production of nanomaterials. The illustrations and references are well done and extensive. I would recommend however, that the authors review and cite Kocot, K. M., McDougall, C. & Degnan, B. M. (2017) (Developing perspectives on molluscan shells, part 1: introduction and molecular biology. Pp. 1-41 in S. Saleuddin & Mukai, S. Physiology of molluscs, a collection of selected reviews. Vol.1. Oakville, Ontario Apple Academic Press), who reached very similar conclusions as the authors' final paragraph. I would also suggest a cautionary statement regarding the MCMCtree57 estimates of divergent times as the dates (but not the overall topology) are highly at odds with the fossil record.

David R. Lindberg

Reviewer #3 (Remarks to the Author):

This manuscript describes the analysis of the genome and biomineralizing transcriptomes of the deep sea scaly-foot snail. This very high quality genome makes an interesting and important contribution the growing number of lophotrochozoan genomes; the generation of sequencing data,

assembly, annotation and primary analysis of the genome is performed in a highly competent manner. It is worth noting that the authors estimate a relatively low number of coding gene models compared to other bilaterians, approx. 17000; this can be at least partially attributed to the high-quality assembly. The authors also report that this snail has fewer unique gene families than other lophotrochozoans, consistent with the relatively low gene number. It would be good to have a statement that this estimate is supported by the mapping of the total transcriptome back to genome. This will give readers confidence that the authors haven't missed novel genes.

In addition to the relatively standard suite of whole genome analyses, the authors focus on the unique biomineralizing features of this gastropod, which is a good idea. They specifically investigate two aspects: the high level of iron in the shell and scales; and the contribution of transcription factor and structural genes to biomineralization. From their analyses they conclude as stated in the Abstract "Comparisons with other lophotrochozoan genomes indicate that the biomineralization toolkit is ancient but with different expression patterns across major lineages. The ability of lophotrochozoan lineages to generate a wide range of hard parts, exemplified by the remarkable morphological disparity in Mollusca, draws on a capacity to dynamically modify the expression and positioning of elements of biomineralisation toolkit across the genome." Specifically, they state that their analyses do not support the hypothesis that morphological novelties are associated with novel genes. This inference differs from the prevailing view about genetic basis of biomaterials including shells and thus a surprise.

Unfortunately, based on the results presented it is difficult to find the data that underpins the authors' conclusions. For example, they show the DMBT1 family has expanded in this snail. Members of this family are predominantly expressed in the mantle. This is certainly an interesting discovery but it is not well enough substantiated to let the authors state "This (gene expansion) likely reflects rapid evolution and expansion of this gene to coordinate protein secretion of scales and shell periostracum". It is important for the authors to look at this gene family more closely, especially given the genes in this family are comprised of SRCR repeats, which are known to evolve rapidly to generate high gene diversity. Given the quality of the genome assembly, it would be good to see a more precise analysis of this gene family (e.g. are there low complexity repeats in some of these genes as found in other 'conserved' biomineral genes) and how each member is expressed in the mantle.

In summary, given biomineralization is main focus of the paper, there needs to be a much more detailed analysis of the mantle- and scale-related transcriptomes and how these genes are organized in the genome. With a detailed analysis the authors conclusions might change. On this point, the authors do not cite many of the papers from which the hypothesis they are refuting. A much higher level of analysis and comparison of these transcriptomes will markedly elevate this study. Given the quality of the genome, this will be the first time where biomineralization genes can be accurately related to expression levels and genome organization. Such a detail analysis is necessary for publication in my opinion. The authors should also consider expanding the current figures, which are skeletal and minimal. This will allow for the inclusion of more detailed results.

Finally, there are statements made throughout the manuscript that are not directly supported by evidence provided (e.g. "Syntenic comparisons reveal both inter- and intra-chromosomal rearrangements between the Scaly-foot Snail and other molluscs (Extended Data Fig. 2-3), and may speculatively contribute to making novel hard parts."). The authors need to temper these types of statements so they are in line with the results presented.

Response to Reviewers' Comments – Sun et al. The Scaly-foot Snail genome and the ancient origin of biomineralised armour

The original comments from the three reviewers are reproduced in full below, with explanation of how the corrections have been implemented. We have incorporated all feedback from the reviewers into the corrected ms. Reviewer comments are **in bold text**, with our response to each point in indented plain text below.

Response to Reviewer #1:

In this study, Sun et al. report the genome of the Scaly-foot Snail *Chrysomallon squamiferum* and explore the biomineralization toolkit of this unusual mollusc with a shell as well as sclerites that may be mineralized with iron sulphide. The genome is of very high quality (chromosome-level scaffolds) and will undoubtedly be a valuable resource to the molluscan genomics research community. The paper is generally well-written (but see minor comments below) and the findings are very interesting with respect to the study of molluscan lophotrochozoan biomineralization. I feel that this paper should be published but that some revision may be necessary.

Many thanks for your time and effort in thoroughly reviewing our manuscript, and we thank you for the overall positive comments. Please find below our point-to-point responses to these concerns, which we hope have been fully addressed in the revised version.

I found the paper to somewhat superficial with respect to some interesting findings. Of course this is somewhat unavoidable given the short format, but I feel that the organization could be improved to emphasize key findings related to the central question of the evolution of molluscan biomineralization raised in the summary.

In the light of this comment, and given that *Nature Communications* allow longer lengths than the original ms in Letter format, we have expanded some parts of the manuscript and rearranged parts of the text in order to improve the organisation.

For example, I question the significance of the reported Hox gene cluster rearrangement to the formation of novel hard parts in this lineage, especially given the absence of even a hypothesized mechanism. Perhaps the reporting of these results and the relevant discussion would best be moved to a supplementary note?

We agree that the discussion of this result was not well-supported, and we have now removed the phrase ‘...and may speculatively contribute to making novel hard parts’.

Additionally, I felt that the discussion of the results of the CAFÉ analysis and the GO analysis, which in my opinion have great potential to address the central question raised in the summary, was underdeveloped.

We have now expanded the manuscript and included discussion of the CAFÉ/GO analyses.

I would encourage the authors to reconsider the organization and ‘real estate’ of the manuscript devoted to the central message(s) of the paper.

Many thanks for this suggestion, and we have now reorganised the entire manuscript and the new format allows us to include additional subheadings within the Results and Discussion. We hope the central messages are easier to follow in the revised version.

I also have a number of other minor suggested edits:

Line 38: “biosynthesis of iron” – iron in an element; animals synthesize iron compounds but not iron itself.

This has been corrected (‘compounds’ added).

Line 43: I would insert “the” before “biomineralization”

Inserted as suggested.

Line 61 and elsewhere: Lophotrochozoa is a clade / grouping so there is no need to put the word “the” before it.

Removed ‘the’ before Lophotrochozoa, vs the vernacular lophotrochozoans.

Line 62: Correct “present” to “presents”

Corrected.

Line 84: This is not the only ONT + Illumina genome for a non-model organism (so this is not a “novel” approach).

Replaced “novel” with “successful”.

Line 88: Correct “among” to “of a”

Corrected as suggested.

Line 89: Something seems wrong here: “...complete ANTP of 11 Hox genes...”

The Hox gene family is a large gene family, and the ANTP is one of the groups. We have expanded this to clarify: *‘the complete Antennapedia Hox gene complex containing 11 Hox genes on chr11’* which should make this clear.

Line 91 and elsewhere (including figures and tables): Patinopecten has been changed to Mizuhopecten.

Corrected throughout the manuscript and figures.

Lines 93-95: Maybe? What would the speculated mechanism be?

As noted above we have now removed this phrase *‘...and may speculatively contribute to making novel hard parts’*.

Line 120: I suggest deleting “the” in “using the samples”

Deleted as suggested.

Lines 142-143: No genes have been reported from the spicules (I think the term sclerite should be used over spicule, by the way), beak, or operculum – these are non-living structures – but the tissues that secrete them.

This has been revised as suggested to

‘... aculiferan mantle and sclerite-secreting tissue, gastropod or bivalve mantle tissue, or tissues secreting radula, beak or operculum’

We agree with your point and throughout the manuscript we now use ‘sclerite-secreting tissue’ or ‘sclerite-secreting epithelium’; and ‘spicule’ has been revised to the more general ‘sclerite’.

Lines 151-152: It is not made clear until this part of the manuscript that the scales of animals from environments lacking iron sulfide have a different composition and the compositional difference is never made clear.

In this expanded revision we have added the relevant information into the introduction:

‘Later, a second population lacking iron sulphide mineralisation was discovered in the Solitaire hydrothermal site characterised by low concentrations of iron (1/58 of that found at Kairei) ...’

Figure 2b: The different parts of this portion of Figure 2 would be easier to interpret if split into more lettered parts with descriptive labels in the figure caption.

The figures have been rearranged and expanded; this is now Figure 4b and the figure caption has been expanded to explain the in situ images in more detail

Figure 3: Please clarify which genes are being compared in part b of Figure 3 in the main text or figure caption.

This is now clarified in the figure caption (still Fig 3b after the rearrangement of figures).

We used all of the genes of each species to search against each other.

Also unless I missed it, 3b is referenced in the main text before 3a.

We have checked the sequence of references to figures and supplementary material in the revised ms.

Methods:

It seems like OrthoMCL was used to infer orthology but the methods only briefly refer to this program. Please clarify that OrthoMCL was used and not just BLASTP. If OrthoMCL was used, the inflation parameter and all-versus-all BLASTP settings used should be provided.

The settings in the OrthoMCL pipeline, which implement the BLASTp, are now included in the Supplementary Information and reference added to point to this supplement.

I would also strongly encourage the authors to make all of the output fasta files generated by OrthoMCL available as supplementary data.

This is now part of our Supplementary Data and also Dryad supplementary dataset.

Line 263: The header inadequately described the text in this section.

We changed the header to “High Molecular Weight DNA Extraction, Quantification, and Quality Control”.

Line 281: Correct “preformed” to “performed”

Corrected.

Line 336: Delete “for”

Deleted as suggested.

Lines 336-337: Why just two rounds of Pilon? Were the pre-Pilon, post-round 1, and post-round 2 Pilon corrections compared?

Yes. The pre-, post-round 1 and post-round 2 Pilon corrections was monitored by Benchmarking Universal Single-Copy Orthologs (BUSCO) analysis. Abbreviation used in BUSCO score: C, complete BUSCOs; S, Complete and single-copy BUSCOs; D, Complete and duplicated BUSCOs; F, Fragmented BUSCOs; M, Missing BUSCOs.

BUSCO score for the pre-Pilon result:

C:93.2% [S:92.0%,D:1.2%],F:0.8%,M:6.0%,n:978

BUSCO score for the post-round 1 Pilon result:

C:96.3% [S:95.1%,D:1.2%],F:0.8%,M:2.9%,n:978

BUSCO score for the post-round 2 Pilon result:

C:96.5% [S:95.3%,D:1.2%],F:0.8%,M:2.7%,n:978

BUSCO score for the post-round 3 Pilon result:

C:96.4% [S:95.2%,D:1.2%],F:0.8%,M:2.8%,n:978

The 1st round of Pilon can significantly increase the BUSCO score. However, the 3rd round of Pilon slightly decreases the BUSCO score, indicating that the Pilon error correction has already reach a plateau and over error-correction may slightly jeopardize the per-base sequencing accuracy. Therefore, we only ran two rounds of Pilon.

Finally, after the HiC scaffolding, BUSCO scores reach: C: 96.6% [S: 95.8%, D: 0.8%], F: 0.9%, M: 2.5% ,n: 978

Line 340: Potentially interesting but definitely not needed: Were the missing BUSCO genes searched for in the genome by BLAST? Are the genes missing in this genome the same ones missing in other gastropod / molluscan genomes?

We report the missing BUSCOs in Supplementary Information. The missing 24 BUSCOs

were searched against the genome by both BLAST and HMMSearch.

The missing BUSCOs were checked in all of the known Mollusca genome. There are four of the them, EOG091G0M09, EOG091G0RWI, EOG091G0Z7J, and EOG091G18B1, which are also missing in all of the Mollusca genome examined so far. Currently, there are no Mollusca or Lophotrochozoan BUSCO databases, so only the Metazoa Database is available.

Lines 356 and 359: I believe the suite is called Juicer and the tool used is called Juicebox.

Revised as suggested on both occasions.

Line 385: Correct “bona-fide” to “bona fide” (italicized).

Corrected as suggested.

Line 413: Correct “divergent” to “divergence”.

Corrected as suggested.

Genome annotations section: Details provided on genome annotation in Maker are inadequate. MinION sequencing section: How many flowcells were sequenced? The authors might also consider providing details on how much data was generated per flowcell in the supplement.

The detail settings of the Maker pipeline are provided in the “Material and Methods”. We used 10 flowcells in total. However, at the beginning when using MinION nanopore sequencing, we came across some issues in purifying high molecular weight DNA from highly mucous tissue with a lot of polysaccharide. That is why we were only able to get around 2-4 Gb per flowcell, initially. Also, the samples were *in situ* fixed by high saturated ammonia sulphate (RNAlater), which is suitable for RNA stabilization but not for high molecular weight DNA preservation. For DNA extraction, we used Genomic DNA Clean & Concentrator (gDCC-10) kit (ZYMO) to purify the DNA and output of each flowcell finally reached over 10 Gb. The details of data generated from each flowcell has now been included into the Supplementary Data.

Supplement: I appreciate that the exact commands used with some of the various bioinformatic tools employed in this study are provided. I would encourage the authors to not hesitate to paste in the exact commands used for more of the tools they employed to help make their assemblies / analyses even more transparent. I would also encourage the authors to make key input files, scripts / commands used, and key intermediate output files from analyses available as supplementary data, especially for the evolutionary analyses that may be of interest to workers studying other areas of animal genomics (e.g., the OrthoMCL analysis result fasta files).

As recommended, all exact commands used in this study have now been provided in the Supplementary Data.

Data availability: More information needs to be provided in this section. It is unclear what will be made publicly available. I am of the opinion that the raw sequencing data, assembled genome, assembled transcriptomes, and all predicted transcripts and protein sequences should be made available at the time of publication of this study.

All raw sequencing data, including Illumina and Nanopore reads, have been deposited in NCBI. The assembled genome, transcriptome, predicted transcripts, proteins were uploaded to Dryad. They will be opened to public when this manuscript is accepted. We have added this information and clarified it in the Data Availability section.

Response to Reviewer #2 (David R. Lindberg):

Sun et al. present an important genomic analysis of one of the most novel taxa found at deep-sea hydrothermal vents. That alone would be worthy of publication. However, the presence of multiple biomineralization sites on the snail as well as the incorporation of iron into the foot scales makes this paper especially important. Their treatment and methodology appears both rigorous and comprehensive. And while of substantial interest to biologists and perhaps paleontologists, the authors also touch on important considerations and insights in bio-engineering and the possible production of nanomaterials. The illustrations and references are well done and extensive.

Many thanks David for taking your time to go over our manuscript, and for your positive words, this is much appreciated.

I would recommend however, that the authors review and cite Kocot, K. M., McDougall, C. & Degnan, B. M. (2017) (Developing perspectives on molluscan shells, part 1: introduction and molecular biology. Pp. 1-41 in S. Saleuddin & Mukai, S. Physiology of molluscs, a collection of selected reviews. Vol.1. Oakville, Ontario Apple Academic Press), who reached very similar conclusions as the authors' final paragraph.

We have now cited Kocot et al. 2017 and thank you for this suggestion.

I would also suggest a cautionary statement regarding the MCMCtree57 estimates of divergent times as the dates (but not the overall topology) are highly at odds with the fossil record.

We have re-run the time-calibrated tree with two further additional calibration points (origin of Pteriomorphia and earliest known Brachiopoda). With these calibration points, our tree agrees much better with the fossil record within Mollusca (e.g., divergence estimate for Pteriomorphia). However, some divergence times, especially the earlier nodes, are still at odds with the fossil record, which is likely attributable to the limitations of the available data (i.e., the low taxon sampling outside of Pteriomorphia). As such, we added a cautionary statement with regards to the interpretation of the time tree shown, as suggested.

Response to Reviewer #3:

This manuscript describes the analysis of the genome and biomineralizing transcriptomes of the deep sea scaly-foot snail. This very high quality genome makes an interesting and important contribution to the growing number of lophotrochozoan genomes; the generation of sequencing data, assembly, annotation and primary analysis of the genome is performed in a highly competent manner.

We are again grateful to the reviewer for going over our manuscript and data thoroughly and for these critical comments, all of which have been incorporated into the revised and expanded ms.

It is worth noting that the authors estimate a relatively low number of coding gene models compared to other bilaterians, approx. 17000; this can be at least partially attributed to the high-quality assembly.

We have now included the information on the relatively low number of predicted coding genes in the genome compared to other bilaterians.

The authors also report that this snail has fewer unique gene families than other lophotrochozoans, consistent with the relatively low gene number. It would be good to have a statement that this estimate is supported by the mapping of the total transcriptome back to genome. This will give readers confidence that the authors haven't missed novel genes.

This has been added in the first subsection of the Results & Discussion: 97.3 % of the *de novo* assembled transcriptome can be mapped to the genome.

In addition to the relatively standard suite of whole genome analyses, the authors focus on the unique biomineralizing features of this gastropod, which is a good idea. They specifically investigate two aspects: the high level of iron in the shell and scales; and the contribution of transcription factor and structural genes to biomineralization.

Many thanks for concurring with our focus of the paper.

From their analyses they conclude as stated in the Abstract “*Comparisons with other lophotrochozoan genomes indicate that the biomineralization toolkit is ancient but with different expression patterns across major lineages. The ability of lophotrochozoan lineages to generate a wide range of hard parts, exemplified by the remarkable morphological disparity in Mollusca, draws on a capacity to dynamically modify the expression and positioning of elements of biomineralisation toolkit across the genome.*” Specifically, they state that their analyses do not support the hypothesis that morphological novelties are associated with novel genes. This inference differs from the prevailing view about genetic basis of biomaterials including shells and thus a surprise.

We have considered this opinion very carefully and discussed this at length among the co-authors. Although we do consider that our results support a main conclusion that novel hard parts in the Scaly-foot Snail originate as a result of dynamic modification of the ancient

biomineralisation toolkit, we also do not believe that novel genes do not play a role. As the reviewer suggested, this nuance was not conveyed in the original text. The presence of unique genes in the Scaly-foot Snail, including novel genes in gene families such as chitin synthase (CS) which are highly expressed in the scale-secreting epithelium, indicates that novel genes do play important roles in the making of novel hard parts. However, their contribution seems to be mainly in the downstream functions and the actual secretion, rather than leading to the actual origin of the novel hard part itself. They are the genes that do the work, but are not the key to novelties in the Scaly-foot Snail.

As such, we have expanded the relevant lines in the conclusions and in the revised discussion, for example:

*‘These integrative analyses did not support the hypothesis that **the origin of morphological novelties is primarily based on novel genes.**’*

*‘**Although novel genes do appear to play important roles in downstream production of the hard parts, the hard parts themselves arise by deploying the conserved biomineralisation toolkit.**’*

Further details on this issue are also contained in the responses below.

Unfortunately, based on the results presented it is difficult to find the data that underpins the authors’ conclusions. For example, they show the DMBT1 family has expanded in this snail. Members of this family are predominantly expressed in the mantle. This is certainly an interesting discovery but it is not well enough substantiated to let the authors state “This (gene expansion) likely reflects rapid evolution and expansion of this gene to coordinate protein secretion of scales and shell periostracum”. It is important for the authors to look at this gene family more closely, especially given the genes in this family are comprised of SRCR repeats, which are known to evolve rapidly to generate high gene diversity. Given the quality of the genome assembly, it would be good to see a more precise analysis of this gene family (e.g. are there low complexity repeats in some of these genes as found in other ‘conserved’ biomineral genes) and how each member is expressed in the mantle.

In the light of this comment, we have carried out further analyses of the DMBT1 paralogues in the Scaly-foot genome. This includes a gene tree, a domain analyses to see how many SRCR repeats and other domains are in each paralogue, and we now show how each paralogue is expressed in the mantle and scale-secreting epithelium (revised ms Figure 4).

In summary, given biomineralization is main focus of the paper, there needs to be a much more detailed analysis of the mantle- and scale-related transcriptomes and how these genes are organized in the genome. With a detailed analysis the authors conclusions might change.

We have significantly expanded the manuscript and included more detailed analyses and better represented data with regards to the mantle and scale-secreting epithelium related transcriptomes and their highly expressed genes.

On this point, the authors do not cite many of the papers from which the hypothesis they are refuting. A much higher level of analysis and comparison of these transcriptomes will markedly elevate this study. Given the quality of the genome, this will be the first time where biomineralization genes can be accurately related to expression levels and genome organization. Such a detail analysis is necessary for publication I my opinion.

We agree, particularly with regard to the need to discuss gene and expression pattern organisation in the genome. In the revised version, we have expanded the figures to illustrate the distribution patterns of novel genes, and made the expression patterns of genes in mantle and scale-secreting epithelium easier to follow. Much of the text added in the expansion of the ms discusses this point.

The authors should also consider expanding the current figures, which are skeletal and minimal. This will allow for the inclusion of more detailed results.

We have expanded the figures as suggested.

Finally, there are statements made throughout the manuscript that are not directly supported by evidence provided (e.g. “*Syntenic comparisons reveal both inter- and intra-chromosomal rearrangements between the Scaly-foot Snail and other molluscs (Extended Data Fig. 2-3), and may speculatively contribute to making novel hard parts.*”). The authors need to temper these types of statements so they are in line with the results presented.

As noted above the revised ms has been carefully edited to capture the nuance of these important points, explain the evidence for our conclusions, and avoid unnecessary speculation.

Removed ‘...and may speculatively contribute to making novel hard parts’.

We have toned down some lines in the concluding paragraph, for example:

*‘These integrative analyses did not support the hypothesis that **the origin of morphological novelties is primarily based on novel genes.**’*

*‘**Although novel genes do appear to play important roles in downstream production of the hard parts, the hard parts themselves arise by deploying the conserved biomineralisation toolkit.**’*

Editorial Note: Reviewer #1 was asked to check the authors' response to the comments of Reviewer #3, who was unavailable to re-review.

REVIEWERS' COMMENTS:

Reviewer #1 (Remarks to the Author):

I commend the authors on this high quality manuscript. All of my major concerns have been addressed and I feel that this manuscript is ready for publication.

Additional comments:

- I feel that excessive emphasis placed on how understanding the genomic toolkit that enables biomineralization is critical to reconstructing the early radiation of major clades.
- Line 52: grammar issue (maybe change ", obscures" to "has obscured") – but this argument is underdeveloped in my opinion
- Line 87: "focussed" typo
- The BUSCO score of the predicted transcripts (in addition to the genome assembly) should be reported somewhere (e.g., lines 115-116?).
- Lines 153-155: The intact Hox cluster has been found in other mollusc genomes. See Wanniger and Wolleson 2019: <https://onlinelibrary.wiley.com/doi/full/10.1111/brv.12439>
- Lines 128-129: this list is already a bit out of date (as is Supplementary Table 2)
- Line 216: t-test?
- Line 368-369: See my comment above. It is not clear how the extreme morphological disparity of molluscan biomineralization explains "challenges in obtaining genomic and phylogenetic resolution"
- Line 502: Something is wrong here.
- Line 505: "bead" should be corrected to "beads"
- Lines 553-554: grammatical issue

Reviewer #1 (Remarks to the Author):

I commend the authors on this high quality manuscript. All of my major concerns have been addressed and I feel that this manuscript is ready for publication.

Many thanks for your positive comment. In the revised version, we have addressed your comments as follows:

Additional comments:

-I feel that excessive emphasis placed on how understanding the genomic toolkit that enables biomineralization is critical to reconstructing the early radiation of major clades.

We understand this is outside the scope of the present analysis but it is important context that motivated our study and is relevant to the interpretation of comparisons with other Lophotrochozoa.

-Line 52: grammar issue (maybe change “, obscures” to “has obscured”) – but this argument is underdeveloped in my opinion

Modified to ‘has obscured’, as suggested.

-Line 87: “focussed” typo

Corrected to ‘focused’

-The BUSCO score of the predicted transcripts (in addition to the genome assembly) should be reported somewhere (e.g., lines 115-116?).

Added the following line to report BUSCO score of genome and transcripts in the main text:

‘... with a metazoan BUSCO (Benchmarking Universal Single-Copy Orthologs) score of 96.6% for the genome assembly and 87.5% for the predicted transcripts’

-Lines 153-155: The intact Hox cluster has been found in other mollusc genomes. See Wanniger and Wolleson 2019: <https://onlinelibrary.wiley.com/doi/full/10.1111/brv.12439>

It is correct that the intact ordered Hox cluster has also been recovered from the *Azumapecten* genome. We have added this: ‘...recovered in one gastropod (*Lottia*), and two bivalves (*Mizuhopecten*, and *Azumapecten farreri*).’ We also cited the reference suggested.

-Lines 128-129: this list is already a bit out of date (as is Supplementary Table 2)

We have added a couple of recently published examples in the main text: ‘(such as *Pomacea canaliculata*, *Mizuhopecten yessoensis*, *Lingula anatina*, *Achatina fulica*, *Sinonovacula constricta*, and *Capitella teleta*: Fig. 3a; Supplementary Table 2)’.

Furthermore, we have updated Supplementary Table 2 with six recently published high-quality lophotrochozoan genomes.

-Line 216: t-test?

This is a paired test in DESeq2, not t-test. This has been clarified.

-Line 368-369: See my comment above. It is not clear how the extreme morphological disparity of molluscan biomineralization explains “challenges in obtaining genomic and phylogenetic resolution”

This has been re-written, we agree the original formation was confusing because two parts of the sentence with different emphasis were combined by a semicolon.

-Line 502: Something is wrong here.

Removed extra '(60 µl)'.

-Line 505: “bead” should be corrected to “beads”

Corrected.

-Lines 553-554: grammatical issue

Sentence revised for clarity.